# Biochemical Hazards during Three Phases of Assisted Reproductive Technology: Repercussions Associated with Epigenesis and Imprinting

**DOI:** 10.3390/ijms23168916

**Published:** 2022-08-10

**Authors:** Yves Menezo, Kay Elder, Patrice Clement, Arthur Clement, Pasquale Patrizio

**Affiliations:** 1Laboratoire Clément, 17 Avenue d’Eylau, 75016 Paris, France; 2Bourn Hall Clinic, Cambridge CB1 0BE, UK; 3Reproductive Endocrinology & Infertility, Miller School of Medicine, University of Miami, Miami, FL 33136, USA

**Keywords:** epigenetics, ART, embryo, homocysteine, methionine, methylation, IVF culture media

## Abstract

Medically assisted reproduction, now considered a routine, successful treatment for infertility worldwide, has produced at least 8 million live births. However, a growing body of evidence is pointing toward an increased incidence of epigenetic/imprinting disorders in the offspring, raising concern that the techniques involved may have an impact on crucial stages of early embryo and fetal development highly vulnerable to epigenetic influence. In this paper, the key role of methylation processes in epigenesis, namely the essential biochemical/metabolic pathways involving folates and one-carbon cycles necessary for correct DNA/histone methylation, is discussed. Furthermore, potential contributors to epigenetics dysregulation during the three phases of assisted reproduction: preparation for and controlled ovarian hyperstimulation (COH); methylation processes during the preimplantation embryo culture stages; the effects of unmetabolized folic acid (UMFA) during embryogenesis on imprinting methyl “tags”, are described. Advances in technology have opened a window into developmental processes that were previously inaccessible to research: it is now clear that ART procedures have the potential to influence DNA methylation in embryonic and fetal life, with an impact on health and disease risk in future generations. Critical re-evaluation of protocols and procedures is now an urgent priority, with a focus on interventions targeted toward improving ART procedures, with special attention to in vitro culture protocols and the effects of excessive folic acid intake.

## 1. Introduction

Assisted reproductive technologies (ARTs), essentially In Vitro Fertilization/Intracytoplasmic Sperm Injection (IVF/ICSI) and the associated embryo culture, are considered safe medical interventions, with some 8 million children born [1,2]. The use of ART has steadily risen in modern society, with 3.3 million cycles and more than 500,000 deliveries per year estimated during 2020. However, an increasing number of reports document concerns in relation to epigenetic/imprinting issues/anomalies in children born as a result of these procedures [3,4,5,6,7,8].

Epigenetic regulation via DNA and histones methylation processes is essential for normal mammalian development. Methylation is a universal biochemical process that covalently adds methyl groups to a variety of molecular targets. DNA repair, protein function, and gene expression involve methylation; it plays a critical role in two major global regulatory mechanisms: epigenesis and imprinting, which involve transcriptional silencing and regulation of imprinted genes. Methylation processes that regulate epigenesis and imprinting determine the characteristics of the specific regulatory processes that differ between male and female genomes. During reproduction, the two genomes that unite to create a new individual are complementary but not equivalent.

In vivo, the maternal oviduct provides unique physiological conditions to ensure correct fertilization and optimal early embryo development, sustaining major waves of epigenetic reprogramming that are crucial for embryonic health. During ART procedures, this critical time is very susceptible and vulnerable to artificial intrusions and environmental changes that can interfere with epigenetic reprogramming, potentially creating epigenetic aberrations that could compromise long-term health in a future generation.

Maternal age is a plausible contributing factor, as the age of patients seeking IVF/ICSI treatment is usually higher than that of the fertile population. The anomalies that have been observed are mainly related to aberrant methylation processes principally affecting DNA (as well as histones) and are strictly related to the ART protocol and not to the etiology of infertility [7,8,9,10]. A recent publication [11] observed that epigenetic problems could reach 5% in the population of Japanese IVF babies, including four pathologies related to imprinting. A review published in 2014 [12] suggested a 3.67-times increased risk of imprinting disorders in children conceived by ART. These observations have been reinforced by many studies in animal models and in humans. A large-scale study comparing cord blood DNA methylation from 962 ART-conceived with 983 naturally conceived newborns revealed widespread differences in DNA methylation between the two cohorts [10]. Overall, a lower methylation was observed across the genome, with differences in 176 known genes, including genes related to growth and neurodevelopment. Methylation differences were found after both fresh and frozen embryo transfers. This study had the unique advantage of including DNA methylation studies of both parents (1956 mothers and 1949 fathers), with 1917 complete “trios” of samples. Methylation differences included hypomethylation at 74% and hypermethylation at 26% of CpG sites in ART-conceived newborns, not found in their parents. After adjusting for age, smoking, BMI, child sex, parity, maternal education, gestational age, birthweight, parental DNA methylation, and cord blood cell composition, the cord blood methylation differences persisted, ruling out effects due to parental methylation and subfertility.

This paper discusses three phases of an ART treatment cycle amenable to epigenetic/imprinting dysregulation: when preparing and performing controlled ovarian stimulation (COH); during the stages of embryo culture; during early embryogenesis for the accumulation of unmetabolized folic acid (UMFA).

Preparation for and controlled ovarian hyperstimulation (COH).Methylation processes during in vitro development and culture of preimplantation embryos.Long-term effects of unmetabolized folic acid (UMFA) during embryogenesis on imprinting “tags”: these are usually sex-specific chemical marks that silence or activate genes. Abnormally high intakes of folic acid (FA) generally used to supplement ART pregnancies lead to accumulation of UMFA, which contributes further to the dysregulation of these three phases.

## 2. Imprinting and Epigenesis

Genomic imprinting is a process whereby either a maternal or a paternally inherited allele is expressed in the offspring. This is achieved mainly via addition of a methyl group to the 5′Carbon residue of a cytosine ring in a CG nucleotide (CpG sites). During early fetal development, when primordial germ cells (PGCs) enter the gonadal ridge, there is a global erasure of methylation marks, including imprinting marks. These are established later, during embryogenesis in males, and at the time of puberty in females, when ovarian cycles are activated. Waves of epigenetic remodeling that take place in the early embryo maintain parental-specific genes, requiring a delicate balance between methylation/demethylation processes. Imprinted genes are essential for the regulation of energy balance between mother and fetus, and they exert differential effects on gene expression during pregnancy: some genes transit through biallelic to monoallelic expression, followed by a return to biallelic expression. This feature of repeated remodeling is a permanent attribute of somatic cells. The kinetics of epigenetic marking are even more complex to determine and to understand; however, epigenetic mechanisms regulate gene activity and cell function and development, and the one-carbon and folates cycles are permanently implicated, both locally and globally.

Epigenetic marking of the male germ line, which is crucial for the transmission of life, takes place within a narrow and critical time window during later stages of embryogenesis. Recent scientific literature has increasingly raised concerns surrounding this critical process, irrespective of the mode of conception. Neurogenic tissues are a particularly sensitive target, and methylation is involved in brain maturation. Defective methylation is known to induce neuropsychiatric and neurovegetative anomalies, and behavioral disorders have been associated with an age-related paternal effect [13,14]. Elevated paternal age is a constant feature in ART.

## 3. Biochemistry of Methylation

An understanding of the biochemical mechanisms that lead to epigenetic modifications is fundamental to appreciate the importance of some compounds of the culture media and environmental factors, such as O_2_ tension and the quality of air, during ART procedures.

S-adenosyl methionine (SAM), the molecule that donates methyl groups (Figure 1), is the universal cofactor for methylation. Methionine Adenosyl Transferase (MAT) catalyzes SAM formation by linking methionine and ATP. Methylation is then carried out by DNA methyltransferase and histone methyltransferase enzymes. After methylation, S-Adenosyl Homocysteine (SAH) and then homocysteine (Hcy) are released. Hcy is a cellular toxin and must be eliminated: SAH and Hcy inhibit methylation, and Hcy also competes with methionine for the same (mainly solute carriers, SLCs) transporter molecules [15] and induces protein conformation anomalies through homocysteinilation.

## 4. Homocysteine Recycling

### Three Different Pathways can Recycle Hcy

Recycling to methionine via interaction with 5-Methyl Tetrahydrofolate (5-MTHF, a component of the folate cycle) and methionine synthase (MS, zinc, and vitamin B12-dependent). Cellular transport of 5-MTHF is carried out by two systems: SLC (solute carrier) 19A1, also known as Reduced Folate Carrier (RFC), which has a poor affinity for folic acid. Folate receptor alpha (FolR1) has a much higher affinity for folic acid; it binds folates and internalizes them by endocytosis [16].Hcy can be recycled by the enzyme Cystathionine Beta Synthase (CBS), leading to the formation of cysteine and then glutathione; all the enzymes involved in glutathione metabolism are present and active in oocytes and preimplantation embryos. This is an important feature, as glutathione protects the epigenetic marking from de-methylation due to undue oxidative stress, thus preserving redox homeostasis [17,18]. The cystathionine beta synthase pathway also generates hydrogen sulfide (H_2_S), which is important in cell signaling and in post-translational modifications [19]. The CBS pathway, however, is poorly expressed in the preimplantation embryo [20] and in the placenta [21,22].A third mechanism, available primarily in the liver, uses betaine homocysteine to recycle homocysteine to methionine: this pathway is also poorly expressed in the human preimplantation embryo and placenta. This pathway is active in the mouse embryo.

## 5. Controlled Ovarian Hyperstimulation, Folates, Methionine, and Homocysteine

Estrogens are potent vasoactive effectors and there is clearly a strong upregulation of estrogen exchange between serum and follicular fluid, as their levels rise during COH (Figure 2). However, folates, methionine, and homocysteine are charged molecules and do not pass freely through the follicular barrier. The final stages of oocyte maturation involve important epigenetic modifications, during which chromosomal regulation and a modification of the epigenetic landscape occur simultaneously. Moreover, in males, the final epigenetic (methyl)-tagging occurs during embryogenesis; in females, this process is activated at/after puberty, during oocyte maturation in preparation for ovulation. DNA methylation is positively correlated with DNA stability [23]. This feature is clearly demonstrated in carriers of MTHFR SNPs (methylene tetrahydrofolate reductase single-nucleotide polymorphisms), who display a very weak capacity to synthesize 5-MTHF, the major cofactor for Hcy recycling [17], potentially resulting in major chromosomal rearrangements/setups such as aneuploidies in their embryos [24].

During follicular growth and maturation, all the biochemical enzymatic pathways implicated in methylation are present in oocytes and are activated [15,20,25], but the energy necessary for their function, and the intermediary metabolites such as folates and methionine, must be supplied to the oocyte and its environment. FA can pass through the follicular barrier to enter the follicular fluid environment of the oocyte, but assays that measure folate in follicular fluid (FF) are inconsistent, and for this reason, FA supplementation appears to increase the levels of FF “folates”, irrespective of the molecules measured [26]. The assays used to measure folate levels have created confusion in the literature, because chemiluminescence assays produce highly ambiguous results as these assays measure all molecules that carry a pteroyl nucleus, which includes 5-MTH, THF (Tetrahydrofolate), DHF (Dihydrofolate), as well as Unmetabolized Folic Acid (UMFA). UMFA has a restricted capacity to enter the folate cycle, as it must first be reduced by dihydrofolate reductase (DHFR), an enzyme with dramatically slow activity [27]. A further block occurs at the MTHFR step, with the end result that UMFA can inhibit the folate cycle [28] (Figure 1).

A crucial requirement for methyl groups is to recycle Hcy as elevated FF Hcy levels may jeopardize oocyte quality [29,30,31,32]. Serum Hcy levels can vary between extremes of 5 and 75 micromoles/liter [26], whether or not patients are taking folic acid supplements. The same variation is observed for levels of serum “folates”. The above paper reported FF levels as slightly lower than serum levels, with variations in FF levels between 1 and 40 µM for Hcy and 1 and 10 µM for folates. Supplementation with FA does not appear to improve intrafollicular levels of “folates” and Hcy as the follicular wall presents a barrier to Hcy, and FA does not appear to improve follicular vascularization. Therefore, any effect of FA on oocyte quality is marginal at best. COH increases the ovarian requirement for methyl groups and folates, and a correlation between low Hcy levels and better oocyte quality has been demonstrated [29,30,31,32]. This indicates that effective vascularization makes a supply of MTHF more readily available, which leads to better recycling of Hcy to methionine.

Hcy and methionine (Met) share the same transporter molecules; the affinity of Hcy is 4 times lower than that for Met [15]; FF that contains an adequate concentration of methionine allows Hcy to be exchanged/released from the oocyte. Excessive supplementation with synthetic folic acid (FA) before IVF, administered, in some cases, at doses of up to 5 mg/day, creates a potential problem involving accumulation of high levels of UMFA (Figure 3, and discussed later) as FA is poorly metabolized to 5-MTHF, especially in carriers of MTHFR SNPs. This means that Hcy recycling to methionine is severely impaired due to the altered methionine synthase activity. Hcy accumulates, as the cystathionine beta synthase pathway is inactive. The folate molecules share the same transporter and receptor molecules, FolR1 and SLC19A1 (solute carrier family 19 member A1), and UMFA can appear in serum at high concentrations. Thus, UMFA alters the transport of the “natural” active metabolite: 5MTHF, ingested in food. A relationship between maternal FA supplementation with >1000 μG has been recently hypothesized as being a hazard for the children: low birth weight and, later on, behavioral problems [33,34]. The findings show that during early female life, environmental exposures to both low and high folate prior to oocyte maturation can compromise oocyte quality, with an adverse effect on offspring of the next generation, partly by altering DNA methylation patterns [35], with unknown/uncertain further impact.

Competition between UMFA and the natural folate (5-MTHF), in combination with inhibition of FA metabolism, impairs the process of methylation by inhibiting Hcy recycling, causing, in turn, elevated Hcy, which is negatively correlated with embryo quality [30,31,32] due to the classical Hcy-related inhibition of methyl transferase [36]. In addition, excessive elevation of estrogens during COH perturbs the entire cycle of 1-CC metabolism, particularly the balance between the cystathionine beta-synthase (CBS) and methionine synthase pathways. This eventually leads to aberrant methylation/imprinting/epigenetic resetting, including impaired histone methylation, during a very critical developmental period [14,17,28].

DNA stability related to methylation is also jeopardized. At the time of fertilization, elevated oocyte Hcy levels and perturbed DNA methylation will lead to an aberrant epigenome that can cause delays during the preimplantation stages of embryo development.

## 6. Early-Stage Embryo Culture, from Fertilization to Genomic Activation

Global methylation of DNA and histones in the sperm nucleus provides a template for the correct spatial and biochemical conformation, necessary to allow rapid transcription factor access to the paternal genome after sperm head nuclear swelling. After fertilization, all regulatory processes are strictly dependent upon maternal reserves of proteins and messenger RNAs (mRNAs) stored during oocyte growth; no major transcription takes place in the newly fertilized egg until the new zygote genome is activated (maternal-to-zygotic transition, MZT). MZT occurs around the 6–8 cell stage of development, approximately 72 h post-sperm penetration. Fertilization creates a metabolic upheaval, with major upregulation of the biochemical pathways necessary for further development.

### 6.1. Cysteine/Cystine Requirement

At a minimum, an increase in the availability of glutathione (GSH) and upregulation of the pentose phosphate pathway (PPP) for NADPH synthesis are both mandatory. NAD(H) is preferentially used in catabolic pathways. PPP and GSH syntheses are directly linked to the folates and the 1-carbon cycles. In general, NADPH is a mandatory co-factor for most of the anabolic pathways that need reduction–oxidation (Red–Ox). Glutathione synthesis facilitates the protection of epigenetic marks [37]. All of the enzymes involved in GSH synthesis and oxidized glutathione (GSSG) reduction are expressed in the human oocyte; GSH is the “control tower” of embryonic redox status and homeostasis [17], and it is also involved in the formation of deoxyribose for nucleic acid synthesis. Thioredoxins (Trxs) are important low-molecular-weight oxido-reductases whose active sites contain cysteine. In summary, cystine, glycine, and glutamic acid, the three components of the glutathione molecule, are crucial in supporting appropriate endogenous GSH synthesis and must be available for the preimplantation embryo.

There is an increased requirement for cysteine, at and immediately after sperm penetration in the oocyte. The oocyte-reduced glutathione necessary for sperm nucleus swelling has been exhausted. As the cystathionine beta synthase pathway is not expressed in the oocyte, correct Red–Ox (i.e., reduced glutathione synthesis) and methylation homeostasis requires an obligatory external supply of cysteine. This is initially provided by follicular fluid, but subsequently must be supported by the in vitro culture medium, strictly reliant upon the presence of this “essential” amino acid as well as the alanine/serine/cysteine transporter 2 (SLC1A5), and the cystine transport activator, the solute carrier family 3 member 1 (SLC 3A1, transports cystine and dibasic and neutral amino acids). Active and passive demethylation processes are known to take place in the pre-implantation embryo, with co-existing demethylation and maintenance of DNA methylation. The early embryo must be protected against undue de-methylation, and cysteine plays an important role in the redox homeostasis required for correct methylation [17]. Errors in methylation are heavily linked to unbalanced oxidative stress.

### 6.2. Methionine Availability Is Crucial

As previously mentioned, methylation requires methionine and ATP to form SAM (S Adenosyl Methionine), the methyl group donor and universal cofactor for methylation. All the enzymes involved in SAM synthesis are present in human, bovine, and mouse oocytes [20]. Methionine adenosyl transferase (MAT) catalyzes SAM formation, linking methionine and ATP. Methylation is then carried out by DNA methyltransferase and histone methyltransferase enzymes. After the targets have been methylated, S-adenosyl homocysteine (SAH) is formed, and homocysteine (HCY) is then released. Experiments with radiolabeled ^35^S methionine have confirmed that all the steps leading to SAM formation and methylation are active in the preimplantation embryo and in humans [15]. Methionine uptake is very active both in oocytes and in preimplantation embryos: this uptake is higher in humans than in the mouse, even taking into consideration the difference in cell size [15]. Hcy is toxic [36] and must be recycled; three potential recycling pathways are outlined above.

Abnormally low/inadequate concentrations of methionine have a profound effect on epigenetic processes, and the dynamics of the methylation process are critical in the early embryo. The zygote genome is thought to undergo rapid DNA demethylation immediately post-fertilization, and this appears to be the case in human IVF embryos. The paternal genome is first rapidly demethylated and then immediately re-methylated, followed by passive demethylation of maternal genes during epigenetic remodeling. The DNA Methyl Transferase 1 (DNMT1) enzyme is highly represented in the oocyte, one of the most abundant mRNAs detected. The level of DNA methylation as measured by the quantity of methylated CpG per embryo does not decrease significantly until the time of the maternal-to-zygotic transition and immediately afterward, contrary to what had previously been reported [38]; it is globally stable up to the time of genomic activation [39,40,41]. In the mouse, SAM synthesis is mandatory for zygotic genome activation where a shortage of methionine will inhibit the completion of cavitation [42]. Methionine/SAM regulates both the (1) Maternal-to-Zygotic transition and (2) blastocyst formation. Bearing in mind that the mouse embryo is the Doxa model for IVF culture quality control via the mouse embryo assay (MEA), methylation must be supported during preimplantation development. Elevated Hcy severely alters mitochondrial function [43,44], providing another reason why it must be eliminated from the environment of the oocyte (the “active mitochondria” are borne by the mother) and embryo. Embryonic transport of methionine is multifactorial and is more efficient than that of other amino acids [45], and the presence of methionine allows the exchange and release of homocysteine.

The cystathionine beta synthase pathways enzyme is allosterically activated and regulated by SAM, augmenting its activity by 2.5× [46]; it increases the formation of cysteine from Hcy, thus facilitating glutathione synthesis: it has a dual protecting role toward methylation in addition to its direct role in redox homeostasis regulation. Hcy and Met exchange is also important; both molecules are transported by similar transport systems, however, with a globally higher affinity for Met [15]. Hcy/Met exchange can be completely blocked by a lack of methionine in the environment of oocytes/early embryos.

### 6.3. Arginine and Polyamines 

Polyamines actively regulate/control methylation, in conjunction with the one-carbon cycle. They bind to DNA and affect gene expression via the interplay between DNA and RNA [47]. Polyamine synthesis is strictly dependent upon a supply of SAM (Figure 4), via SAM decarboxylase and arginine. Methionine deficiency will alter the formation of polyamines, introducing another defect into the regulatory systems surrounding gene expression.

### 6.4. The Folates

Folates are a mandatory requirement for Hcy recycling in the human embryo, but the “folates “present indifferent culture media are difficult to ascertain. ISM1 had folic acid but this medium is no longer available. The type of folate used also deserves consideration as folate metabolism is jeopardized in carriers of single-nucleotide polymorphisms affecting the methylene tetrahydrofolate reductase enzyme (MTHFR SNPs), to the extent that DNA methylation reactions cannot be correctly processed, as clearly demonstrated [27]. Although the oocyte contains all the mRNA’s coding for FA metabolism in a “normal” population, the potential problem in those affected by MTHFR SNPs could be bypassed by adding 5-MTHF instead of FA to the medium. Adding 5-MTHF to culture media should be more beneficial for maintaining both correct epigenetic marking and adequate DNA stability [23,24].

## 7. Impact on ART 

In the context of the biochemistry outlined above, current in vitro culture systems seem to be unfavorable (Figure 5). If we accept the Mouse Embryo Assay (MEA) as the standard for the definition/validation of IVF culture media QC, it is notable that support of these key pathways is absent. This has been clearly demonstrated in two very important papers [48,49]. The current concept of “essential amino acid toxicity” [50,51,52,53] contributes an additional “methylation/epigenetic regulation” burden for the human embryo in vitro as the three “basic pillars”, methionine, cystine, and arginine (and 5-MTHF), are omitted [54] (Table 1).

The culture system itself does not allow appropriate methylation/epigenetic marking.An increase in the rate/speed of embryo development, sometimes considered as a standard of embryo quality, is due to a defective imprint methylation procedure [49]. The timescale necessary for 2 biochemical processes, translation of the stored mRNAs and epigenetic marking, must be respected.

Once again, let us bear in mind that the MEA is the basis for consensus regarding the quality of culture media. Methionine concentration affects global DNA methylation in general, as well as the methylation of specific imprinted genes and, as a result, alters the quality of IVP (In Vitro-Produced) embryos [55]. This observation is not anecdotal: in cattle, it can lead to the large offspring syndrome (LOS) [56,57]. Unsurprisingly, methionine imbalance and related DNA methylation anomalies affect more than 1500 genes involved in protein catabolism and autophagy [55]. Culture media in current use also spontaneously generate oxidative stress [58]: when incubated alone, under IVF culture conditions, without an embryo, they generate and release reactive oxygen species (ROS), a further significant observation that should be considered in relation to media composition.

## 8. Salient Points Surrounding the Importance of Methylation Processes during In Vitro Culture

Three amino acids play a crucial role: methionine, cystine, and arginine. These amino acids have been labeled/considered to be “essential amino acids”, since the early days of cell culture, as mammalian cells in culture are not able to synthetize these amino acids. It is obvious that this has nothing in common with mammalian embryo culture.

The published literature has suggested that the “essential amino acids”, when added in culture media, may be “toxic” for early development [50,51,52,53] because they delay the rate of development. This assertion/concept is highly controversial/questionable in the context of early embryo development. This represents the “crux of the problem”. To protect epigenetic/imprinting marking, the three amino acids mentioned above should not be removed, but should instead be supplied in concentrations that take into consideration transport molecule affinities. Methionine will allow an exchange and a release of homocysteine. Elevated Hcy severely alters mitochondrial function [43,44,59]. Methionine must be present and Hcy must be eliminated. The risk of elevated ammonia concentration in the medium, proposed to support this pseudo-“essential amino acids toxicity”, is not a valid concern, as both (NH_4_) HCO_3_ and (NH_4_)_2_CO_3_ are unstable compounds. Ammonia can be released in the gas flow or eliminated by transamination with the formation of alanine [18,60]. Neither the presence of essential amino acids nor the presence of ammonium ions affect the development of feline embryos [61].

An effort must be made to improve the culture media composition, eliminating statements that are not based upon physicochemical principles. A solution may not be straightforward or simple; however, mimicking tubal fluid composition might be an advantageous starting point. Oviductal fluid contains glycine, glutamine/glutamic acid, cystine (for glutathione synthesis), methionine, and folates.

Protection against oxidative stress and maintenance of correct epigenetic marking are both of equal importance and concern. The addition of external antioxidant compounds may help transitorily but this cannot replace the mandatory endogenous synthesis of glutathione. It cannot be added in culture media, as its transfer through biological membranes is weak/poorly active.

## 9. High Doses of FA during Pregnancy, Unmetabolized Folic Acid (UMFA), and the Impact on Embryo Health

In the event that a preimplantation embryo manages to successfully overcome the hurdles of fertilization and suboptimal culture media (because deficient in the “essential” amino acids necessary for its epigenetic health), it then faces a further challenge. As mentioned earlier, nutritional fortification +/− high doses of folic acid supplements have led to a syndrome associated with unmetabolized folic acid (UMFA). In countries where foodstuff fortification is mandatory, UMFA has been identified in placenta, cord blood, and circulating in infants’ blood [62,63,64]: this suggests that UMFA has been present throughout the period of gestation. Prescribing FA doses of at least 1 mg/day prior to starting an IVF treatment and continuing during the first trimester is now standard practice in some countries. The fact that UMFA now appears to be ubiquitous, even in countries where fortification is not implemented [65], creates a significant dilemma: whether folic acid is added “everywhere”, or patients assume it on their own, the optimal/necessary dose of FA to be prescribed during the first months of pregnancy has become impossible to define, unless circulating UMFA is first estimated before starting an IVF treatment.

Elevated Hcy is by no means benign [66] and has also been recognized as a risk factor for vascular dementia, Alzheimer’s disease, and schizophrenia [66,67,68]. Any dysfunction or impairment of the 1-CC and folates cycles may affect brain development and function; this is also the case for folate receptor defects that induce folate deficiency in the brain [68]. Folate receptor and transporter molecules are present and active in the placenta [20,21] and are an important contributing factor that must be taken into consideration: all of the “folates”, including FA, share common primary receptors, folate receptor R1 (FolR1) and transporters (SLC19A1, solute carrier A1), as well as a secondary pH-dependent transporter: proton-coupled folate transporter PCFT/SLC46A1, primarily involved in intestinal folate absorption. Folate receptor α (FolR) has a higher affinity for the nonreduced folates (FA) vs. 5-MTHF (reduced folate), and the opposite is the case for the solute carrier 19A1 (SLC19A1) [21,22]. As mentioned earlier, dihydrofolate reductase, the enzyme involved in the first step on integration of FA in the folate cycle, has limited activity in humans. An excess of FA may induce a Michaelis and Menten feedback effect, further reducing the capacity to metabolize folic acid. Receptors/transporters may become fully saturated, creating a local block in the availability of 5-MTHF for homocysteine recycling; this will impede all the steps of the folates cycles between tetrahydrofolate and 5-MTHF. Carriers of MTHFR SNPs may have limited access to 5-MTHF, which can lead to a “folate malabsorption syndrome”. Scientific observations linking this syndrome to an excess of UMFA now provide a warning that the possibility of this malabsorption should be avoided. Somatic cell methylation processes in general can be affected, as well as specific effects on maturation of neurogenic tissue and epigenetic resetting in male germinal cells [69].

It is a well-known fact that pregnant patients take vitamin supplements that are higher than recommended doses, and this may have an impact on fetal brain development [34]. Circulating UMFA has been detected even in countries where foodstuff fortification is not implemented [65], and FA now appears to be present “everywhere” in body fluids, as are endocrine disruptor chemicals (EDCs). Significantly, EDCs are present in maternal body fluids, in the placenta, and in embryos [70]. They exert a now well-known potent effect on methylation/epigenetic processes [36,71,72], and can act as negative catalytic cofactors during embryo development. Reduced IQ and impaired neurodevelopment have been reported [72,73], and this could be partly attributed to negative effects of EDCs on metabolic pathways [37,71]. It is clearly now imperative to exert caution, considering the differing roles and effects of 5-MTHF vs. FA, their doses and timing of administration, as well as the effects of a potential MTHFR SNP genetic background.

## 10. Discussion and Conclusions

Medically assisted procreation and IVF and ICSI are considered a safe and reliable therapeutic approach to help infertile patients to achieve parenthood. Millions of healthy live births are solid proof that the majority of IVF and ICSI treatment cycles are safe. However, an increasing number of reports now suggest that the manipulation of gametes and embryos may induce aberrations in epigenesis and imprinting. What is of greater concern is that it may take an entire generation before such epigenetic aberrations may become noticeable.

A biochemical analysis and breakdown of the relevant physiology and metabolic pathways points to three phases of an ART process that provide food for thought and further discussion.

### 10.1. Controlled Ovarian Stimulation in the Presence of Abnormally High Doses of FA

Epigenetic resetting in the female gamete occurs during this period of oocyte maturation. A local supply of methyl groups and folate is imperative to ensure that methylation can be appropriately managed. Homocysteine elevation in follicular fluid has been observed; however, the negative impact of Hcy can be reversed: Hcy can be either recycled or removed from the early embryo via exchange with methionine. This requires an appropriate culture medium that contains the essential sulfur amino acids methionine and cystine, allowing antioxidant protection via glutathione synthesis. Farm animal embryo transfer programs that apply superovulation with uterine flushing and transfer of blastocysts that have developed in vivo, confirm that the negative impact of Hcy accumulation after COH can be reversed. A parallel situation in humans can be seen when “uncontrolled ovarian stimulation” followed by artificial insemination has led to the development of “multiplets” (quadruplets and quintuplets).

### 10.2. Duration of In Vitro Culture

This phase presents a major hazard for embryo development. Culture media in current use do not have the capacity to maintain appropriate methylation processes, even in the mouse, where it needs to be correct if the mouse embryo assay (MEA) is accepted as a standard for quality control. The notion of eliminating essential amino acids from media, especially methionine (and arginine), results in further metabolic stress that accelerates protein catabolism and affects autophagy and apoptosis associated with mitochondrial stress [43,44,45,46,47,48,49,50,51,52,53,54,55,59]; this may be associated with or independent of elevated Hcy. Due to limited/absent cystathionine beta synthase activity in the early human embryo, cysteine is mandatory, its status transitioning to “essential”. The concentration of these two sulfur amino acids should probably be increased, considering the high capacity for transport of methionine, which may impede cellular entry of other amino acids [45]. It is now time to “seriously” reconsider the definition of human IVF media, [74], and the relevance of the MEA to human embryo culture. Concern regarding elevated circulating Hcy and its negative impact in conceptus quality and fertility have recently led the “Rotterdam periconception cohort” to recommend the “*routine analysis of homocysteine levels in preconceptional and pregnant women and their partners*” [75].

Two concepts must be seriously re-evaluated: “the essential amino acids toxicity, and the, related or not, speed of development”. The spatial and temporal biochemical processes have to be respected, especially the interplay between translation of the mRNAs stored during oocyte growth [76] and the methylation processes and its protection, against oxidative stress, via the endogenous synthesis of glutathione, a mandatory pillar of redox homeostasis.

### 10.3. Folic Acid Supplementation

This third phase is more difficult to capture. Folic acid supplementation with 0.4 to 0.8 mg/day is recommended prior to ART treatment, continued throughout the first trimester. However, a landscape of confusion and turmoil is created with the addition of foodstuff fortification programs as well as access to numerous different types of commercial complement preparations marketed to “enhance fertility” or provide support for these precious pregnancies. The definition of an appropriate dose may vary from one patient to another, and classical fluorescence-based assays used to determine serum folate levels do not distinguish between unmetabolized folic acid, 5-MTHF, and other folate compounds that have a pteroylglutamic core. The issues associated with MTHFR SNPs, and T677T, must not be overlooked. Estimation of real and relevant serum folate levels requires more sophisticated assays, including mass spectrometry and chromatography; however, these tests are expensive and time-consuming.

In conclusion, the impact of high doses of folic acid with resulting systemic circulating UMFA now requires re-evaluation, with particular attention to the context of male germ cell epigenetic setting as well as general epigenetic setting during embryo development [69]. Attitude changes to overcome the perception that ‘more is better’ have become an urgent priority.

## Figures and Tables

**Figure 1 ijms-23-08916-f001:**
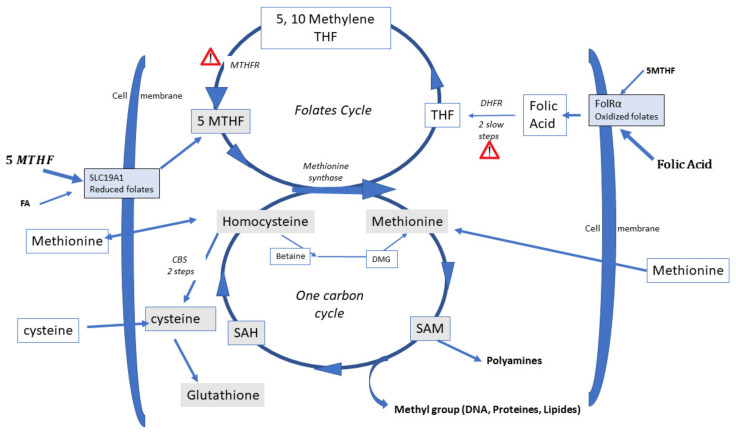
Folates and One-Carbon Cycles. There are 2 slow steps driven by DHFR, dihydrofolate reductase, allowing the entry of folic acid (FA) into the folate cycle. FA binds preferentially to folate receptor alpha (Folα) and is then internalized. Then, the MTHFR SNPs (methylene tetrahydrofolate reductase single-nucleotide polymorphisms) can alter the capacity to generate MTHF (MethylTetraHydrofolate), the only active compound interacting with the Methionine Synthase (MS), to regenerate Methionine from Homocysteine (Hcy). 5 MTHF is preferentially transported into the cells via the transporter SLC19A1 (reduced folates transporter, Solute carrier). Homocysteine can be recycled in 2 steps to form cystathionine, then cysteine: this pathway is not expressed in the early preimplantation embryo and is weak/absent in placenta. Another pathway using betaine allows the regeneration of homocysteine to methionine and forms dimethylglycine. This pathway BHMT (Betaine Homocysteine Transferase: BHMT; DMG: Dimethyl Glycine) is also poorly active in the early embryo and placenta. S-Adenosyl Methionine (SAM) releases a methyl group on the target molecules and forms S-adenosyl homocysteine. Exclamation mark symbols indicate two crucial steps that regulate entry of FA into the folates cycle and its activity.

**Figure 2 ijms-23-08916-f002:**
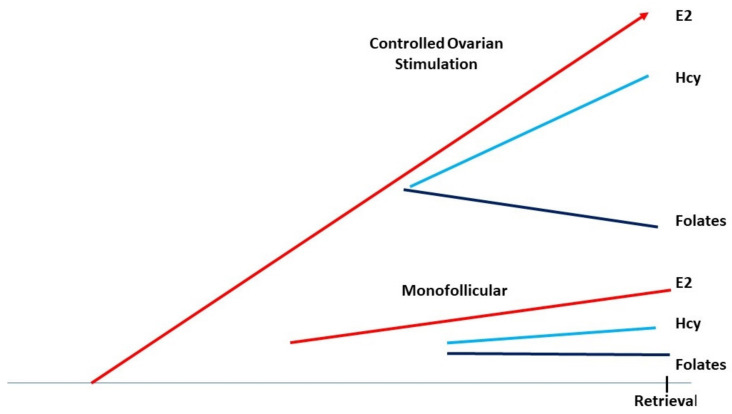
The evolution of circulating estrogens Hcy and 5-MTHF levels in follicles and oocytes during ovarian stimulation.

**Figure 3 ijms-23-08916-f003:**
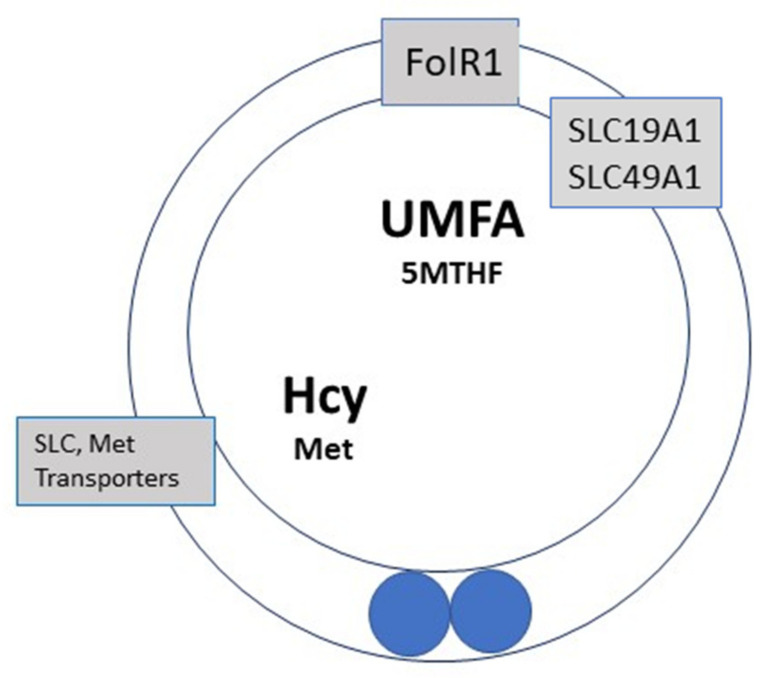
Metabolic status during the time of fertilization and in vitro culture of a zygote in an environment containing excess folic acid. Folate and methionine are absent from fertilization media; Hcy level is high and Met concentration is low. The concentration of UMFA is much higher than the concentration of 5-MTHF; UFMA saturates the 1-CC, which impairs recycling of Hcy. The situation is even more complicated in MTHFR SNP (methylene tetrahydrofolate reductase single-nucleotide polymorphism) carriers. SLC: Solute carrier transporter, FolR1: Folate receptor1, SLC19A1 and SLC49A1 folates transporters.

**Figure 4 ijms-23-08916-f004:**
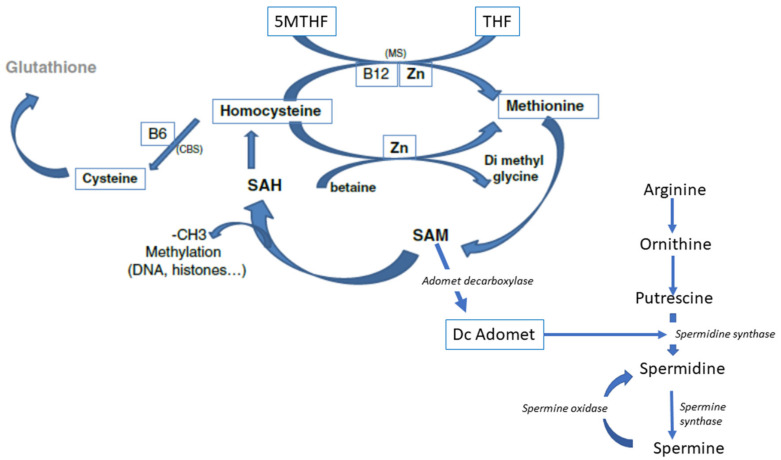
Polyamine synthesis is linked to the availability of SAM and supply of arginine.

**Figure 5 ijms-23-08916-f005:**
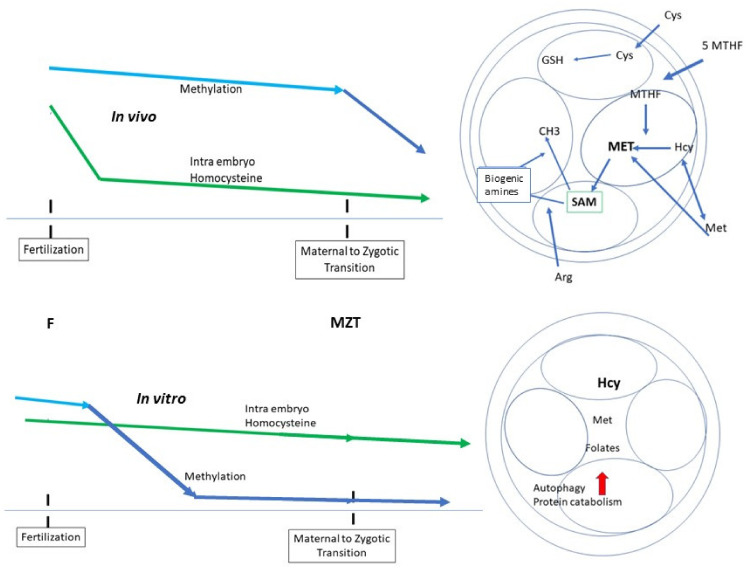
Embryo metabolism in vivo or in media containing methionine, arginine, cystine, and folates (upper), and in IVF media containing no or a very weak concentration of essential amino acids, especially methionine [17,18,48,55,58] and no folates.

**Table 1 ijms-23-08916-t001:** The status of 3 essential amino acids in the culture media designed for early embryo culture from fertilization to genomic activation (D0 to D3). * This medium has replaced ISM1, which had Arg 138 µM, Met 89 µM, and Cys 42 µM.

	Vitrolife	Sage	COOK	In Vitro Care	Origio	IVF Online	Cooper	Irvine
Medium	G1	QACM	SICM	IVC1	Seq Fert/Cleav *	Global	Universal IVF	CSC
Cys	0	0	2	0	0	52	0	46
Met	0	0	4	0	0	51	0	53
Arg	0	0	25	0	0	328	0	281

Adapted from *Morbeck* et al. *Fertil. Steril* 2014 [54] and informations collected from the culture media dealer at https://fertility.coopersurgical.com/art-media-products/. All the values are expressed in µM.

## Data Availability

Not applicable.

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
