# Peer review of "Biochemical Hazards during Three Phases of Assisted Reproductive Technology: Repercussions Associated with Epigenesis and Imprinting"

_ijms, 2022, doi:10.3390/ijms23168916_

Round 1

Reviewer 1 Report

See attached file

Author Response

Rev  1 :

We have modified all the specific points, as requested by the reviewer (underlined). We do agree that it is a mix of molecular biology and intermediate metabolism, a usually neglected parameter

The two major points are

*the “larga manu” use of folic acid. Readers have to be aware of the complicated metabolism of folic acid, especially for MTHFR SNP carriers

* The problem of culture media and the “erratic” concept of “essential amino acids toxicity”, adding a supplementary burden/hurdle to the embryo.

We have explained a little bit more the methylation before entering in the “hard” of folate

We have added a table with the composition of the culture media (Morbeck 2014). It is surprising that the concept still exists after the “terrible “papers of Market Velker et al. ref 48, 49. Nobody seems to care and the “officials” like NIH should be aware of the risks.

For “folates”: their metabolism has to be understood: poor entry in the folates cycle (DHFR, Bailey and Ayling); then risk of blockade at MTHFR

Reviewer 2 Report

The authors put together a very comprehensive and succinct review regarding the epigenetic changes associated with ART births compared to non-ART normal births and how several metabolic factors like UMFA might contribute to these differences. The review covers a lot about underlying epigenetic and imprinting disorders in ART. Would it be possible to elaborate more about the invitro culture conditions and how this can be improved in the future? What are the authors perspective about how to address the culture condition with metabolic supplements? The authors can try to reorganize the schematic figures to make them more illustrative. There is a typo in line 60 (a period is missing).

Author Response

we have tried to be more “understandable”. For the culture media, there are miracle compounds to add. The media have to “push” the endogenous synthesis of glutathione Addition of antioxidants in culture media can help marginally. The best example is given by the rescue of oxidized methionine.: It can be only managed by thioredoxins with Glutathione: no other way!!!!... intermediate metabolism. Another link between imprinting and protection against ox stress

A table of the culture media composition is given

Reviewer 3 Report

This is a very comprehensive review on Assisted Reproductive Technology (ART) associated with epigenetics/imprinting. A review for wide range of audience with different level of understanding on DNA/histone methylation, Menezo et al. discussed what are the known features of DNA/histone methylation for ART, described the molecular mechanisms, roles, and interplays of methylases and demethylases, explained the Homocysteine recycling pathway, and discussed the impact on ART. The paper is unambiguous without any concerns.

Author Response

No special request: 

No specific concern.